# Emotional Needs in the Face of Climate Change and Barriers for Pro-Environmental Behaviour in Dutch Young Adults: A Qualitative Exploration

**DOI:** 10.3390/ijerph23010076

**Published:** 2026-01-05

**Authors:** Valesca S. M. Venhof, Bertus F. Jeronimus

**Affiliations:** 1System Earth Science, Maastricht University, 5900 AA Venlo, The Netherlands; 2Faculty of Behavioural and Social Sciences, Developmental Psychology, Groningen University, 9712 TS Groningen, The Netherlands

**Keywords:** environmental change, mental well-being, youth, psychological distress, sustainable behaviour

## Abstract

**Highlights:**

**Public health relevance—How does this work relate to a public health issue?**
This work addresses the growing mental health burden associated with climate change among young adults, a group increasingly recognized as vulnerable to climate-related distress.By simultaneously examining both climate-related emotional responses and the emotional and psychological needs of young adults, the study provides insights for designing interventions that support mental wellbeing, foster resilience, and promote adaptive coping in the context of climate change.

**Public health significance—Why is this work of significance to public health?**
The study provides empirical, context-specific insights into climate-related emotions and emotional needs related to climate change, among Dutch young adults, which can inform more locally informed approaches to prevention, communication, and support programmes.Understanding barriers to pro-environmental behaviour informs public health interventions that can reduce climate-related health risks, promote sustainable behaviours, and support the mental wellbeing of young adults experiencing eco-anxiety or feelings of helplessness.

**Public health implications—What are the key implications or messages for practitioners, policy makers and/or researchers in public health?**
The study shows that climate change can elicit stress, anxiety, powerlessness, and hopelessness among young adults, underscoring the need for a comprehensive, multi-level public health response that simultaneously addresses emotional needs, structural barriers, and opportunities for meaningful engagement.Lowering barriers to pro-environmental behaviour and fostering supportive environments that enable sustainable action among young adults may enhance wellbeing and strengthen their sense of agency. Effective public health policy should address knowledge gaps, motivational factors, social norms, structural barriers, and institutional responsibilities to foster sustained engagement and wellbeing.

**Abstract:**

Rapid climate change and its anticipated impacts trigger significant worry and distress among vulnerable groups, including young adults. Little is known about how Dutch young adults experience and cope with climate change within their specific social and environmental context. This study examines Dutch young people’s emotional responses to climate change, their perceived emotional and psychological needs arising from these experiences, and the barriers they encounter in engaging in pro-environmental behaviour, with the aim of informing public health strategies to better support and empower this vulnerable group. Data were drawn from a large online survey among a representative sample of 1006 Dutch young adults (16–35 years; 51% women). The questionnaire included fixed-answer sections assessing emotional responses to climate change, as well as two open-ended questions exploring participants’ perceptions of their emotional and psychological needs related to climate change and the barriers they perceive to pro-environmental behaviour. Descriptive statistics were used for the fixed-response items, and thematic analysis was applied to the open-ended responses. Many Dutch young adults reported worry and sadness about climate change and its impacts, with approximately one third experiencing feelings of powerlessness. A large percentage of respondents attributed responsibility to large companies, and nearly half indicated that they still had hope for the future. One third (31%) felt that nothing could make them feel better about climate change, and another third (36%) reported to experience no climate-related emotions. Key emotional needs included more action at personal, community, and governmental levels, and more motivating positive news. Almost half (46%) of young adults said they already lived sustainably, while perceived barriers to pro-environmental behaviour were mainly financial (21%), knowledge-related (8%), and time-related (7%). This exploratory study highlights key practical and emotional barriers to pro-environmental behaviour reported by Dutch young adults 16–35, who expressed diverse emotional needs while coping with climate change. The findings underscore the need for a multi-level public health response to climate-related emotions, that simultaneously addresses emotional needs, structural barriers, and opportunities for meaningful engagement. Lowering barriers to pro-environmental behaviour and fostering supportive environments that enable sustainable action among young adults may enhance wellbeing and strengthen their sense of agency. Public health supports this by reducing barriers to pro-environmental behaviour in young adults, through targeted support, clear information, and enabling social and structural conditions that promote wellbeing and sustained engagement.

## 1. Introduction

Our world is undergoing rapid environmental change, most prominently through anthropogenic climate change and its interconnected challenges, including biodiversity loss, extreme weather events, and growing ecological degradation (e.g., [1,2]). In addition to their significant ecological and material consequences, these accelerating changes have profound social and psychological impact. A growing body of research shows that climate change can negatively affect mental wellbeing by intensifying stress, anxiety, and feelings of loss and uncertainty [3,4]. Climate change affects mental health both directly and indirectly, giving rise to a range of emotional, cognitive, and behavioural responses (e.g., [3,4,5]). Direct impacts include psychological distress following climate-related disasters such as floods, wildfires, heatwaves, and storms, which are associated with increased levels of anxiety, depression, post-traumatic stress symptoms, and prolonged grief (e.g., [6]). Indirect impacts arise through anticipatory stress, perceived loss of environmental stability, and concerns about future living conditions, contributing to emotions such as chronic worry, helplessness, anger, sadness, and moral distress; often discussed under the umbrella term of climate anxiety (e.g., [7,8,9]).

Climate stress responses are shaped by multiple processes including exposure to climate-related risk, uncertainty about future socio-environmental conditions, perceived lack of control, and concerns about the adequacy of political and institutional responses (e.g., [10,11,12]). Climate-related emotional stress can elicit adaptive coping strategies such as information-seeking, pro-environmental behaviour, collective engagement, and meaning-focused coping, versus more maladaptive or avoidant strategies, such as denial, emotional disengagement, minimisation of risk, or withdrawal [13,14]. Action-oriented and meaning-focused coping seem to foster a sense of agency and purpose [15,16]. Prolonged climate-related distress in the absence of perceived pathways for effective action may increase the risk of emotional exhaustion, disengagement, and avoidance [7,8]. Furthermore, chronic stress is associated with heightened risk of a wide range of mental and somatic health problems [17,18].

Groups that are particularly vulnerable to climate-related stress and mental health impacts are young people, Indigenous communities and others who have strong emotional or livelihood-related ties to their natural environment, those who live in climate-exposed regions, and individuals with pre-existing mental health conditions [19,20,21]. We zoom into the experiences and vulnerabilities of young adults (aged 18–35) who just as children and adolescents proved to be particularly vulnerable for mental health problems, also due to their climate change awareness [8,22,23]. Youth are busy forming their identity and future planning and realize they (simultaneously) face an unprecedented (sense of) threat regarding their life course and prospects [8,15,24]. Young adults (ages 16–35) are at a critical life stage in which habits, values, and coping strategies are formed, shaping long-term environmental engagement [24].

A recent ten-nation global survey of 10,000 youth aged 16–25 showed ~83% experienced a profound loss of faith in society’s stewardship of the ongoing ecological crisis [8]. The present study was aimed to examine how Dutch young adults’ responded to climate change in terms of their emotions and perceived emotional and psychological needs and barriers to pro-environmental behaviour. Climate-related emotions are an important mental health factor and climate change stress can elicit feelings of anxiety, hopelessness, and powerlessness, particularly among young adults, affecting their wellbeing and resilience [8,9,15,25]. Understanding young adults’ emotional needs and coping strategies can guide interventions that prevent mental overload and strengthen resilience, both at the individual and population level. Barriers to pro-environmental behaviour may further increase stress and feelings of powerlessness, highlighting the importance of addressing structural and motivational obstacles to more sustainable lives. This study aims to generate insights that can inform public health strategies to support mental wellbeing, resilience, and adaptive engagement in the context of climate change. 

An emerging body of research indicates that young people worldwide experience climate-related emotions and environmental distress, including high levels of worry and grief, alongsidechronic anxiety, sadness, and helplessness [8,9,15]. At the same time, these emotions frequently coexisted with positive emotions such as hope and increased political or collective engagement (e.g., [8]). A recent cross-national European study found substantially higher levels of climate worry in Portugal, Austria and Slovenia and lower levels in Sweden, Iceland and Italy. In pooled analyses across 11 countries, climate worry was associated with an increased risk of clinically significant anxiety, although it was not linked to depression or sleep disturbance, and the strength of associations varied between countries [26].

We studied climate change stress in the Netherlands where high population density coincides with increasing exposure to climate risks such as sea-level rise, and ongoing pressures from diffuse and cumulative stressors (e.g., pollution, biodiversity loss, urbanisation). These factors intersect with other influences on mental wellbeing, such as school and personal problems [27,28]. 

In this paper we focus on young Dutch adults of Generations Z (born 1997–2007, thus age 16–26 in this 2023 study) and Millennials (born 1988–1996 aged 27–35), who were already under considerable strain due to the societal measures in response to the COVID-19 pandemic [27]. The specific role of climate change and Dutch young people’s perspectives on climate change and environmental degradation in shaping their psychological wellbeing has received little attention. Emerging evidence indicates that young people living in Europe experience significant worries and distress related to climate change and show substantial variation in emotional and psychological responses to climate change across European countries [8,23,26]. This underscores the need for focused research on young adults and highlights the importance of national-level studies to complement and contextualise international findings. Locally informed research can support the development of public health strategies that strengthen resilience and empower vulnerable groups, such as young adults in the Netherlands, as national and cultural contexts shape both experiences of climate-related distress and the resources available for coping.

This research is part of a large nationally representative online survey, conducted in 2023, which included 1006 Dutch young adults aged 16–35, stratified by gender, age, education level, and province [29,30]. The overall aim of the large survey, which consisted of distinct subsets of questionnaires and has been analysed and reported in three independent studies, was to gain deeper insight into environmental distress, climate-related emotions, and climate change-related perceptions and psychological needs among young adults living in the Netherlands. The results presented here come from the final study, which focuses specifically on Dutch young adults’ emotional responses to climate change, their perceived emotional and psychological needs arising from these experiences, and the barriers they face in engaging in pro-environmental behaviour. Unlike the two previous studies, which used a purely quantitative design, we now use a mixed-methods approach and enrich findings on fixed response items with results from open-ended questions (see Section 2). While the quantitative survey data capture the prevalence and distribution of climate-related emotions within a nationally representative sample, the thematic analysis of the qualitative data provides depth and context.

We now first summarize the findings from the two previous studies based on the same nationally representative survey (Section 1.1), then contextualize these results in a review of emotional and psychological needs arising from climate change (Section 1.2) and the barriers young adults face in engaging in pro-environmental behaviour (Section 1.3), before outlining the theoretical framework (Section 1.4) and presenting the aim of the present study (Section 1.5).

### 1.1. Summary of Previous Survey Findings

In our first study (published elsewhere: [29]) we examined environmental distress, which we defined as the psychological stress and negative emotional responses arising from perceived environmental change and deterioration more broadly, including phenomena such as solastalgia—the distress experienced when one’s home environment is degraded or altered. Unlike climate change-specific emotions (such as anxiety) these are broader constructs that capture responses to a wider range of environmental stressors from noise, pollution, and loss of nature. We also investigated how factors such as place attachment, sense of control, trust, and personality moderated the psychological stress responses of 1006 Dutch young adults [29]. Participants (16–35 years) most commonly reported stress in their natural environment related to noise (~22%), while loss of natural areas (~20%) and heat (~18%), were perceived to be most threatening. Approximately one fifth of 1006 Dutch young adults indicated that environmental distress limited their enjoyment of life, while nearly a quarter expressed worries about the future. Feelings of powerlessness (~27%) and low trust in governments were also reported. The findings revealed a significant portion of young adults experiencing environmental distress, while others appeared indifferent [29]. Although the survey items differed, international data indicate that considerably higher proportions of young people reported that climate change negatively affected their daily life and functioning (≈45%), perceived the future as frightening (≈75%), and believed that people were failing to take care of the planet (≈83%; [8]). These findings highlight substantial cross-national variation, which we discuss in more detail below.

Our second study (published elsewhere [30] focused on climate change-related distress and denial in the same representative sample of Dutch young adults. Previous large international surveys indicate that most young people aged 16–25 are very or extremely worried about climate change (≈59%), with over half reporting emotions such as fear, sadness, anxiety, anger, powerlessness, or guilt [8]. Complementary qualitative studies among children and adolescents aged 7–18 have documented feelings of sadness, hopelessness, and guilt in response to climate change itself, alongside anger and frustration directed at the perceived apathy and inaction of adults and previous generations [31]. Together, these studies point to a recurring theme of ethical betrayal and moral injustice, which has been shown to negatively affect young people’s mental health (see also [23,32]). Building on this body of work, we used latent profile analysis to identify six distinct emotional profiles among the young participants: ‘burdened worriers, unburdened worriers, climate change deniers, sceptic worriers, ‘Not in My BackYard (NIMBY)’s, and conflicted sceptics. These profiles capture patterns of distress, denial, and coping within Dutch young adults (16–35), and provide a nuanced understanding of how they responded emotionally to climate change [29,30]. Climate change distress and denial coexisted in complex patterns, for example, high-distress profiles were associated with hope and proactive coping, whereas denial-heavy profiles were linked to fatalism, lower institutional trust, and reduced engagement with the natural world. Minimal differences were observed across gender, age, income, or living environment categories/brackets, and no indication of consistent differences by education. These results underscore the nuanced and complex ways in which Dutch young adults responded emotionally to climate change and suggest the need for interventions tailored to specific profiles of emotional distress [30].

Together, both studies showed that Dutch youth aged 16–35 experienced substantial environmental (Study 1) and climate-related (Study 2) distress, characterized by diverse emotional responses ranging from worry and feelings of powerlessness to denial, and that these responses were shaped by individual, social, and contextual factors. The current study (Study 3) builds on these previous works by shifting the focus to a descriptive examination of young adults’ emotional responses to climate change. Whereas the second study primarily identified distinct profiles of climate change-related distress and denial, Study 3 explores the broader spectrum of emotions experienced by Dutch young adults. To better inform policy and practice on how these emotions may be alleviated and supported, and to identify factors that may hinder engagement in pro-environmental behaviour, the survey included open-ended questions addressing participants’ perceived emotional and psychological needs related to climate change, as well as the barriers they perceive to pro-environmental action. In the following Section 1.2 and Section 1.3, we first summarize recent research on emotional and psychological needs and barriers to sustainable behaviour to provide a framework for interpreting aforementioned open-ended responses.

### 1.2. Emotional Needs Related to Climate Change

International research into psychological and emotional needs in the face of climate change highlights that young adults need hope, empowerment, and constructive engagement to counter feelings of helplessness and climate anxiety [9,15]. Adolescents and emerging adults (aged 18–25) seek to be heard and taken seriously by older adults, policymakers, and institutions [8], as well as opportunities to engage in collective and meaningful climate action that fosters agency and belonging [33,34]. Moreover, supportive social networks and open dialogue with peers, teachers, and parents were described as crucial for processing climate-related emotions [15,16]. European reviews and large surveys also showed that young people reported high levels of climate-related distress and that responses from youth services, schools and health systems are needed to provide psychosocial support and opportunities for meaningful engagement [6,8]. These findings suggest that addressing environmental distress among young people requires more than individual-level coping strategies and demand relational, societal, and policy-level responses that acknowledge their concerns and enable constructive participation, a theme to which we return in the Section 4 below.

### 1.3. Barriers to Pro-Environmental Behaviour

The interplay between climate-related emotions, pro-environmental behaviour, and mental wellbeing is complex and remains relatively underexplored [35]. Global awareness of environmental issues grew rapidly in the 20th century. A 2024 United Nations Development Programme (UNDP) survey on climate action—conducted among over 73,000 people in 77 countries, representing 87% of the world’s population—revealed a strong global consensus: 80% of adults called for more decisive government action to address climate change [36]. Globally, only a small share of adults (7%) believed that their country should not transition at all, while a large majority (79%) wanted wealthier nations to help poorer countries adapt [36]. Public perceptions of environmental issues are also shifting in the Netherlands. Almost half (45%) of Dutch young adults (ages 18–24) viewed environmental pollution as a serious problem in 2019, although only 11% considered it a major issue [37]. In recent years the Eurobarometer showed that 91% of 15–24-year-olds believe that tackling climate change can improve their health and well-being [38]. Moreover, most young adults (60%) acknowledge the need for a more climate-conscious lifestyle [39]. However, research highlights a significant gap between these pro-environmental attitudes and actual behaviour [40]—for example, more than 60% of 18–25-year-olds reported flying in the past year [39], few follow a plant-based diet [39], and many continue to purchase new clothing [41].

Perspectives on individual responsibility for pro-environmental behaviour vary across regions [40]. For example, a survey of 1000 American adults showed half of them believed that businesses should promote sustainable practices, and only a third saw sustainability as a personal responsibility [42]. Environmental awareness and pro-environmental behaviours are also influenced by individual characteristics such as personality traits and perceptions, and cultural and regional contexts [14,43,44]. Analysing data from the 2016/2017 European Social Survey, Martin et al. [45] found that adolescents and young adults who felt personally responsible for addressing climate change reported greater happiness and life satisfaction. In contrast, frequent worry about climate change was associated with lower well-being [45], see also [30]. Despite growing awareness, barriers to pro-environmental behaviour persist in everyday settings—at home, school, work, and in public spaces [40]. These barriers span a range of factors, including informational gaps, psychological perceptions, social norms, and cultural influences [46,47]. In Europe, sustainable consumption behaviours are linked to higher levels of environmental knowledge and risk perception [48]. Moreover, feeling empowered to act sustainably can help reduce environmental distress and foster collective engagement (e.g., [15]). The United Nations Environment Programme’s (UNEP) Global Survey on Sustainable Lifestyles highlights that young adults around the world aspire to contribute to sustainability but often look for clearer guidance on how to do so effectively [49,50]. In general, young people are more likely to adopt sustainable behaviours when they observe others doing the same, demonstrating the influence of social proof [50].

### 1.4. Theoretical Framework

There is a growing body of research demonstrating that climate-related emotions play a complex and sometimes ambivalent role in shaping young people’s responses to climate change, influencing wellbeing, engagement, and behavioural intentions in different ways [25,35,51,52]. Building on this literature, the present study examines emotional responses to climate change within a nationally representative sample of Dutch young adults, alongside their perceived emotional and psychological needs arising from these experiences, and the barriers they encounter in engaging in pro-environmental behaviour. By combining these dimensions in a large, representative sample of Dutch young adults, this study contributes novel insights into how cognitive, behavioural, and emotional factors intersect in shaping responses to climate change. This study builds on stress theories that describe how our body responds to internal or external demands that exceed the resource capacities of the person or social system, a transactional perspective in which health and successful aging reflect our ability to adapt and effectively respond to the dynamic challenges of being alive [17,18], and we focus on climate change as the chronic stressor that forces us to adapt to maintain equilibrium or homeostasis with our surroundings. Our key questions pertain to how Dutch young adults think they are handling climate change. 

Most research on climate-related emotions and sustainable behaviour has relied on quantitative survey instruments with closed, theory-driven items, such as standardized scales for pro-environmental intentions or climate anxiety [53,54,55]. While questionnaire approaches are valuable for identifying population patterns and prevalences, they constrain participants to predefined categories which limits serendipity. A key contribution of the present study lies in its mixed-methods design, in which we combine quantitative data on emotional responses to climate change with qualitative insights into the emotional and psychological needs young adults articulate, as well as the barriers they perceive to pro-environmental behaviour. Our survey data captures the prevalence and distribution of climate-related emotions within our nationally representative sample, our thematic analysis provides depth and context to these proportions and may help understand *how* these emotions shape perception and coping behaviour in everyday life. Together, these complementary approaches allow for a more nuanced understanding of how emotional experiences, perceived needs, and behavioural constraints intersect.

Although this study is exploratory in nature, our research questions are informed by the Theory of Planned Behaviour (TPB; [56]) which posits that behaviour is influenced by individuals’ (a) attitudes toward the behaviour, (b) perceived behavioural control, and (c) subjective norms, which thus are the hypothesized key personal and social factors that facilitate or hinder sustainable actions [56,57]. We explore the construct of perceived behavioural control in this paper. This explorative study of emotional needs in the face of climate change builds on perspectives from environmental psychology, which emphasize the role of emotions, coping, and psychological resilience in relation to climate change (e.g., [58,59]). Rather than applying these frameworks in a formal, hypothesis-driven manner, we use them as guiding perspectives to situate our exploratory findings within broader theoretical debates.

### 1.5. Aim of This Research

The aim of this study is to examine Dutch young adults’ (16–35) emotional responses to climate change, alongside the emotional and psychological needs they perceive in relation to these experiences, and the barriers they encounter to engaging in pro-environmental behaviour. Understanding climate-related emotional responses, perceived needs, and barriers within the Dutch socio-cultural context is essential for developing effective, locally informed, and culturally appropriate public health interventions and policies that promote mental wellbeing and resilience, empower young adults, and meaningfully engage them as active participants in addressing climate-related challenges.

## 2. Materials and Methods

Data were drawn from a large nationally representative online survey, comprising three analytically distinct studies, of which the present paper reports the third study (see also [29,30]). The sample included 1006 Dutch young adults aged 16–35 years (51% women). The questionnaire-based study was conducted between 27 February 2023 and 9 March 2023, and included demographic variables, personality traits (BFI-10; [60]), self-perceived mental and physical health, environmental distress (Study 1, [29], climate change-related emotions (Study 2, [30]), Study 3 (current)), and two open-ended questions (Study 3, this paper). Detailed descriptions of the survey design, questionnaire development, and results are provided in Venhof et al. [29] and Reitsema et al. [30]. The full questionnaire and its English translation are available online [61].

This study focuses on the section of the questionnaire that combines fixed-response items assessing emotional responses to climate change with two open-ended questions exploring participants’ perceived emotional and psychological needs related to climate change and the barriers they perceive to pro-environmental behaviour. Emotional responses to climate change were assessed using a 0–100 Visual Analogue Scale (VAS; 0 = ‘no emotion’, 100 = ‘very strong emotion’), on which participants rated the intensity of 12 climate-related emotions experienced over the past four weeks. To enable comparison between climate-related emotions and general emotional states, four emotions—loneliness, anxiety, depressed mood, and stress—were eventually selected from the set of 12, as these same emotions had also been assessed earlier in the questionnaire using the same VAS in a general (non-climate-specific) context (see [29]). Participants therefore rated these four emotions twice: once in relation to their general emotional state and once specifically in relation to climate change.

The survey also included 20 self-designed items by the first author, presented on a 5-point Likert scale, asking about emotional responses to climate change. These items were based on, and adapted with permission from, a qualitative study on eco-emotions [62] and adjusted to the Dutch context. Specifically, the items covered eco-anxiety (6 items), eco-guilt (8 items), and eco-coping including pro-environmental behaviour (6 items). Part of the data has been used in a latent profile analysis, conducted by an independent researcher, on climate change distress and denial (Study 2, [30]). However, the descriptive results on the broader emotional responses to climate change, which we present here, were not presented in that paper. 

At the end of the survey, two open-ended questions (see Table 1) were presented, asking about perceived emotional and psychological needs related to climate change and the barriers they perceive to pro-environmental behaviour. The questions were developed by the authors. Participants were asked to provide a maximum of three written responses in a text box for each question; skipping was not allowed. For both questions, a fixed response option was available if the question was not relevant to the participant. By integrating qualitative insights from these two open-ended questions, with quantitative survey data on the emotional responses to climate change, this study aims to provide a more nuanced interpretation of young adults’ emotional responses to climate change, their perceived needs, and the barriers to sustainable behaviour.

Participants were recruited through Flycatcher Internet Research, a panel certified for quality (ISO 20252) and environmental standards (ISO 14001). Eligibility criteria included being 16–35 years old, residing in the Netherlands, Dutch literacy, and providing informed consent. The final sample consisted of 1006 Dutch young adults aged 16–35 (M = 25.9, [SD = 5.6]), of whom 51% identified as women. Stratified sampling was applied based on gender, age, education level, and province, and data collection followed the ‘Golden Standard’ calibration instrument [63] to ensure national representativeness. Participants provided informed consent and completed digital questionnaires that included built-in measures to minimize missing or inconsistent data. Ethical approval was granted by Maastricht University (FHML-REC/2022/130), and participants received a small financial incentive. This study was funded by the Heymans Data Collection Fund and BFJ received support from the research project ‘Stress in Action’ (www.stress-in-action.nl; accessed on 20 February 2023; financially supported by the Dutch Research Council and the Dutch Ministry of Education, Culture and Science; NWO gravitation grant number 024.005.010); neither of which influenced the study design nor outcomes.

### Data Analysis

Participant data was transferred to IBM SPSS Statistics (Version 28.0) by an independent researcher from Flycatcher Internet Research, including the responses to the open-ended questions displayed in Table 1. 

For the four emotional responses to climate change (loneliness, anxiety, depressed mood, and stress) that were rated on a VAS scale slider bar (0–100), the descriptive statistics such as the mean, SD, median and interquartile range (IQR) were estimated. The median indicates the central tendency of the VAS scores, while the IQR reflects the spread of the middle 50% of responses, which we use as a robust measure of variability, less affected by extreme values. Because VAS scores were not normally distributed, we conducted non-parametric analyses. Paired comparisons between general and climate-specific VAS scores for each emotion were performed using Wilcoxon signed-rank tests (SPSS v30). Effect sizes were calculated using the formula r = Z/√N [64]. To assess the degree of association between general and climate-specific emotions, Spearman’s rank-order correlations were computed for each emotion. Statistical significance was set at α = 0.05, and we applied Bonferroni correction to account for alpha inflation due to multiple comparisons between the four emotions (general vs. climate specific) in IBM SPSS Statistics Version 28.0. Comparing climate-specific ratings of anxiety, stress, depressed mood, and loneliness with their general counterparts allows us to examine whether climate-related emotions reflect a distinct emotional response rather than a mere expression of general emotional distress.

The 20 items assessing emotional responses to climate change, measured on a 5-point Likert scale, were analysed descriptively using SPSS version 30. For the analysis, response categories were collapsed, with 1 and 2 combined as ‘(totally) disagree,’ 3 as ‘neutral,’ and 4 and 5 combined as ‘(totally) agree.’

Responses to the open-ended questions were exported from SPSS v28 to Microsoft Excel (MS Excel Professional Plus, 2021) by the first author. Descriptive statistics (frequencies) were first calculated for the fixed ‘not applicable’ response options (B and C, Table 1). Open-text responses (A, Table 1), for which participants could provide up to three answers, were organized in a separate Excel file. The first author then conducted a thematic analysis following Braun and Clarke [65], including: familiarization with the data, generating initial codes, searching for overarching themes, reviewing and refining themes, and defining and naming the final themes. To enhance the robustness of the analysis, the thematic coding was performed twice at separate time points. Responses deemed irrelevant or off topic were documented and grouped into a separate ‘irrelevant’ category. These qualitative data complement the quantitative findings, providing context and depth to the two open-ended questions on participants’ perceived emotional needs and barriers to pro-environmental behaviour. Comparing climate-related emotions with broader emotional tendencies within the same sample allows for a more nuanced understanding of perceived climate threat.

## 3. Results

The sociodemographic details of the sample of Dutch young adults are published in full detail in Venhof et al. [29]. In short, 1261 respondents were included, of which 32 provided no informed consent or were not living in the Netherlands, another 159 did not complete the questionnaire for unknown reasons, and another 64 were removed by an independent researcher from Flycatcher. The final dataset consisted of a national representative sample (age; gender; education level; province of residence) of 1006 respondents. The average age was 25.9 [SD = 5.6]) years, with 51% women. More details of the sample are provided in Venhof et al. [29].

Participants reported lower climate-specific anxiety (Median = 11, IQR = 3–37) than general anxiety (Median = 25, IQR = 10–51; *Z* = −11.34, *p* < 0.001, *r* = −0.36, indicating a moderate effect size [66,67]. Similar patterns were observed for depressed mood (Median difference = 11; *Z* = −16.04, *p* < 0.001, *r* = −0.51, indicating a large effect size [64]), and stress (Median difference = 20; *Z* = −18.63, *p* < 0.001, *r* = −0.59, reflecting a large effect size [64], loneliness (Median difference = 5; *Z* = −13.50, *p* < 0.001, *r* = −0.43, indicating a medium-to-large effect size [64].

Spearman’s rank-order correlations (rₛ) revealed significant positive associations between general and climate-related emotions, with moderate effect sizes for anxiety (rₛ = 0.39, *p* < 0.001), depression (rₛ = 0.31, *p* < 0.001), stress (rₛ = 0.26, *p* < 0.001), and loneliness (rₛ = 0.35, *p* < 0.001). These findings indicate that neighbouring climate-specific emotions were only moderately correlated with general emotional states, suggesting that climate-related emotional experiences represent a distinct but connected emotional domain. Detailed results for the four climate-related responses, anxiety, depression, stress and loneliness, are presented in Table 2.

Emotional responses to climate change were assessed with additional items using a 5-point Likert scale (1 = ‘completely agree’ to 5 = ‘completely disagree’). Most respondents (57%) expressed worry about the future of coming generations, such as their children, or reported feeling sadness for people and animals affected by climate change (55%). Approximately one third reported feelings of powerlessness (31% for item 6 and 35% for item 12, see Table 2) and most (61%) also believed that large companies were responsible and should do more to address climate change. Only 18% agreed with the statement that there is no more hope or that it is too late, while almost half (49.2%) disagreed, which indicated that many Dutch young adults still see reasons for hope. Only one fifth (~20%) found support on climate-related topics from peers or organisations, while about 26% tended to avoid the topic altogether and focused mainly on activities they enjoyed. See Table 2 for an overview of these results.

In terms of pro-environmental behaviour, nearly half of the participants (46%) stated that they were already making changes to their lifestyle, and one third (36%) felt they were personally doing enough. A smaller group (26%) believed they were not doing enough to live more sustainably. Additionally, 42% focused primarily on actions within their personal sphere of influence, and 23% reported confronting others about sustainable living. 

All 1006 Dutch young adults responded to the two open-ended questions exploring their emotional needs related to climate change and their perceived barriers to pro-environmental behaviour. For the question on emotional needs, about two-thirds of the participants (673/1006) selected one of the fixed response categories (B/C). One third (31%, 307/1006) indicated that ‘nothing could make them feel better about climate change,’ while a similar proportion (36%, 366/1006) reported having ‘no feelings about climate change.’

One third of respondents (33%, 333/1006) provided at least one open-ended response; 9% (88/1006) gave two answers, and 4% (38/1006) provided the maximum of three. Of the total 459 open responses, 13 were deemed irrelevant as they did not pertain to the question. Although the answers were highly diverse, several clear themes emerged (see Figure 1).

A subset of young adults expressed a desire for more action at the personal and community level (10%, 101/1006), for example: *‘we have to start acting now, starting yesterday.’* Others emphasised the need for stronger action by industry, government, and/or other countries (8%, 75/1006), such as *‘I need action from the government, instead of words,’ ‘I need better decisions from the government, they make a mess of it,’* and *‘the big countries in the world need to act.’*

Another group highlighted the need for accessible information, including reliable news and practical guidance (5%, 50/1006): *‘I need honest information”* and *‘I need clear information on what I can do.’* A smaller group stressed the importance of broader societal acknowledgement of the problem (3%, 26/1006), with statements such as *‘our society should take the problem more seriously’* and *‘more people should become aware of it.’*

Several participants expressed a wish for more positive news, focusing on improvements and progress (6%, 55/1006): *‘show us what works, positive stories,’ ‘more positive stories in the news, not only the negative,’* and *‘I need perspectives for the future.’* Finally, a small group (2%, 20/1006) indicated they needed *‘possibilities for everyone to contribute,’* for example through *‘a national plan in which civilians can also participate’* or *‘provide possibilities for individuals to do something.’*

Not all answers could be categorised within these themes; and 115 responses were grouped into the category ‘other’ (see Figure 1). Within this “other” group, twelve participants mentioned the need for more insight into the impact of their own actions, and five indicated a desire for more conversations about the topic. Many responses concerned practical needs, such as *‘more electrical cars,’ ‘solar panels,’ ‘living more sustainably,’* financial support and subsidies (N = 22), or general *‘support’* (N = 6). A small number of youth (N = 8) denied climate change altogether, stating it was *‘all exaggerated,’ ‘negativity,’ ‘just a natural phenomenon,’* or that we *‘should not make such a big problem of it.’* Another three participants responded that we should *‘just accept it.’*

In response to the question on perceived barriers to pro-environmental behaviour, 42% of participants (426/1006) provided input in the open-text fields, while the majority (58%, 580/1006) selected an option from the fixed response categories (answer ‘B’/‘C’). Among those who used the open-text option, many gave multiple responses: 26% of the total sample (257/1006) provided two answers, and 12% (125/1006) provided three answers. In total, 807 open responses were collected, of which approximately 7% (60/1006) were judged irrelevant. Most of these irrelevant responses were still related to pro-environmental behaviour, such as *‘flying less’* and *‘using less water’* but did not address the specific question of *‘what holds you back to live more sustainably’*.

Nearly half of the participants stated that they *‘were already doing what they could’* (46%, 463/1006). One in eight (13%, 129/1006) indicated that they were not personally or individually responsible for acting more pro-environmentally. An additional 3% of youth (29/1006) gave similar responses in the open-text fields, identifying others—such as industry, government, or other countries—as primarily responsible for sustainable action: *‘big companies need to act first’* and *‘the government needs to act’*. In line with this, 4% (36/1006) argued that individual impact is very limited, for example: *‘one person has no influence’* and *‘I cannot change it as one individual’*, with related themes including *‘I have no power/young people are not listened to’* (N = 14/1006) and *‘my age/I’m too young’* (N = 5/1006). All perceived barriers to pro-environmental behaviour are summarized in Figure 2.

The most frequently mentioned barriers in the open-text fields concerned financial constraints (21%, 208/1006), with examples such as *‘alternatives such as the train are expensive’*, *‘other choices cost a lot of money’*, and *‘I have limited financial means’*. Knowledge-related barriers were mentioned by 8% (83/1006), for example: *‘current information is contradictory’*, *‘inconsistent’*, or *‘I lack clear and consistent information on what I can do’*. A lack of time was also frequently cited (7%, 74/1006), as in *‘I’m too busy with other things’* or *‘I’m too busy with my job’*. Many respondents reported that suitable alternatives were lacking, or lower in terms of quality, or *‘less fun’* (~4%, N = 42/1006), or they felt that pro-environmental behaviour required *‘too much effort’* (~4%, N = 44/1006).

Another important theme concerned behavioural change, reflected in statements such as *‘I don’t want to give up my (comfortable) lifestyle’* (N = 39), *‘taking the first step is difficult’* (N = 9), and *‘I have no motivation’* (N = 28). Other respondents perceived barriers placed on pro-climate behaviour driven by external forces (i.e., for which others were responsible), such as *‘others also do nothing’* (N = 16) and *‘my social environment’* (N = 32): *‘people in my social environment will see me in a negative way’*, or *‘I live together with not like-minded people’*.

Smaller themes related to personal wellbeing included answers such as *‘my physical wellbeing’*, *‘my health’*, *‘worries’*, *‘stress’*, *‘shame’*, and *‘energy levels’* (N = 21); governmental decisions and *‘unclear’* regulations or politics (N = 18); denial (N = 12), such as *‘climate change is a lie’*, *‘is it really as bad as they say’*, and *‘it is a natural phenomenon’*; and housing issues (N = 8), such as *‘I live too remote’* and *‘I live in an old property’*. Additionally, some young adults (3%) stated that sustainable behaviour was *‘pointless’*, indicating that *‘it has no use’* and *‘it is already too late’*. Some answers could not be categorised within the aforementioned themes. The category ‘other’ (3%, 27/1006) included a variety of miscellaneous responses on perceived barriers to pro-environmental behaviour, such as *‘I’m not afraid’*, and *‘climate activists’.*

## 4. Discussion

With this mixed-method study we deepened our understanding of Dutch young adults’ (16–35) emotional responses to climate change, together with the emotional and psychological needs they perceive in relation to these experiences, and the barriers they encounter to engaging in pro-environmental behaviour. The number, diversity, and quality of responses demonstrated strong participant engagement. Compared to their general emotional states, Dutch young adults (aged 16–35) reported lower intensity levels of anxiety, depressed mood, stress, and loneliness when these emotions were assessed specifically in relation to climate change. The small to moderate correlation between general and climate-specific emotions (*r*ₛ = 0.258–0.389, all *p* < 0.001) indicated overlap (R^2^ = 0.07 to 0.15). This suggests that while individuals with higher general emotional distress also tend to report stronger climate-related emotions, these emotions are largely independent quantities. Hence, climate-related emotions seem to represent a specific emotional domain (with terms like ‘eco-anxiety,’ ‘eco-anger,’ ‘climate distress,’ or ‘solastalgia’) within a broader emotional landscape, more complex than merely reflecting general distress. This conclusion aligns with the small to moderate meta-analytic correlations between eco-anxiety and depression and general anxiety/depression measures (roughly *r* = 0.20 to 0.40, see [66]) and differences in the factor structures of climate emotions versus general distress, and in their external correlates (e.g., risk perception, trust in government, emotion regulation strategies [67]). 

In this discussion, we integrate the qualitative insights with the quantitative findings to provide a more nuanced understanding of young adults’ emotional responses, perceived needs, and barriers to pro-environmental behaviour. Based on this integrated perspective, we derive preliminary insights and practical implications for public health, which are further elaborated in Section 4.2.

### 4.1. Climate-Related Emotional Needs and Mixed Emotions

Dutch young adults described a complex emotional landscape in response to climate change, characterised by moral concern, ambivalent hope, perceived powerlessness, and diverse forms of engagement and disengagement. Quantitative results indicated that a majority expressed worry about the future of coming generations, including their children (57%), and sadness for the people and animals expected to suffer as a result of climate change (55%). These findings highlight that climate-related emotions among young adults are not limited to personal anxiety; they seem to reflect a *need for meaning, moral acknowledgement, and validation*. This aligns with literature describing climate distress as existential and value-based (e.g., [8,9,15,25]).

At the same time, almost half of the participants (49%) in our study reported hope for the future, while nearly one in five (18%) expressed feelings of hopelessness, and one third indicated that climate change made them feel powerless. This coexistence of hope and despair suggests that hope among young adults may be fragile or conditional, dependent on the availability of credible pathways for action and meaningful engagement (cf. [68]). This emotional tension, characteristic of climate-related distress (e.g., [7,51,69]), underscores the need for *credible pathways to hope* through realistic and meaningful opportunities for agency. Previous work also indicated that hope and righteous anger can coexist and can motivate collective action [70]. We need to learn when and how this happens.

Our quantitative data further revealed diverse perspectives on who was responsible for the climate crisis: most participants (61%) believed that large companies should play a more active role, and many (46%) reported taking personal steps towards more sustainable lifestyles. Nonetheless, one quarter (26%) indicated they were failing to live sustainably, and over a third (36%) believed they had already done enough to mitigate climate change. Together, these findings suggest that feelings of powerlessness and disengagement may arise from both emotional distress and perceived structural constraints regarding who is responsible for climate action [52,71,72], underscoring a *need for collective action and structural agency*. 

The qualitative data from the open-ended questions on emotional needs provided a complementary perspective that enriched the quantitative findings and clarified what young adults thought was required for them to cope with climate-related distress. We identified several themes that we now outline. About one third of respondents (31%) indicated that ‘nothing could make them feel better’ about climate change, suggesting a perceived lack of agency and a sense of hopelessness that may increase the likelihood of emotional distancing or disengagement as a defensive response when emotions become overwhelming [71,72,73]. A similar proportion (36%) reported having ‘no particular feelings’ about these problems. Rather than simple indifference, this pattern may also reflect emotional numbing or avoidance, which have been described in the literature as strategies through which individuals protect themselves from climate-related distress [15,73,74]. 

Among the one third of participants who provided open-ended responses, the following clear themes emerged:

Desire for personal and community action: Approximately 10% of Dutch young adults expressed a need for immediate action at the personal or community level, reflecting a wish to regain agency and see tangible engagement (‘we have to start acting now, starting yesterday’).

Call for collective and structural action: Around 8% of Dutch young adults emphasised the need for stronger efforts from industry, government, or other countries, highlighting the perception that meaningful climate action depends on systemic change rather than individual effort.

Access to reliable information and guidance: Five percent of Dutch young adults requested more accessible and trustworthy information, including practical guidance on how to act (‘I need clear information on what I can do’).

Positive framing and progress-oriented news: Six percent of Dutch young adults stressed the importance of seeing solutions and progress, suggesting that positive messaging may help sustain hope and engagement.

Opportunities for participation: A smaller subgroup (2%) of Dutch young adults indicated a need for avenues through which everyone could contribute, such as national plans enabling civilian participation.

Other responses included practical support (e.g., subsidies, sustainable technologies), the desire for dialogue, and, in a minority, outright denial or acceptance of climate change. These qualitative insights align closely with the quantitative patterns of engagement and disengagement, underscoring the heterogeneity of emotional responses and needs among Dutch young adults. The findings suggest that supporting young adults’ mental wellbeing and resilience in the context of climate change requires tailored support and differentiated strategies that acknowledge varying emotional positions and capacities for engagement. This includes not only individual-level emotional support, but also collective, structural, and relational conditions that foster agency, meaning, and sustained engagement (in line with [3,9,25]). Importantly, the qualitative data make explicit the needs for action, information, positive framing, and participatory opportunities, thereby highlighting actionable pathways for interventions. 

In a broader European context, these results are consistent with pan-European studies documenting high climate concern alongside substantial variation in emotional engagement, perceived agency, and coping responses across and within countries (e.g., [8,25]). European research similarly reports the coexistence of worry, hope, powerlessness, and avoidance, as well as the central role of structural responsibility and perceived fairness in shaping emotional wellbeing and climate engagement. A recent review of 69 studies (1995–2025) on climate-related emotions in children and young people concluded that beyond personal vulnerability to climate impacts, social and political factors are pivotal in shaping youth mental health outcomes in the climate crisis [23].

### 4.2. Barriers to Pro-Environmental Behaviour

Dutch young adults displayed moderate to high individual engagement in pro-environmental behaviour, as many reported lifestyle changes (46%) or felt that they already did enough (36%). Yet, a notable proportion (26%) felt they were not doing enough, and fewer engaged in collective or social actions (23%, see also Table 2). Exploring the barriers which Dutch young adults perceived to engage in pro-environmental behaviour, several broader themes emerged (see Figure 2). First, a large proportion of respondents indicated *structural and systemic barriers*, especially financial constraints (21% (208/1006) reported lack of money) or lack of (appealing or accessible) alternatives (4%) and housing issues (1%), next to governmental decisions and politics (2%). Additionally, one in six (~13% (129/1006)) adults felt that addressing climate change was ‘not their responsibility,’ while 3% (29/1006) explicitly identified industry, government, or other countries as primarily responsible. These responses suggest that many young adults perceive meaningful climate action as dependent on systemic changes rather than individual effort, consistent with research on structural responsibility and perceived behavioural efficacy (e.g., [72,73]). Our results also suggest that many Dutch young adults judge mildly. This contrasts with previous international research showing that a majority—up to two-thirds—of young people experience the inaction of governments and previous generations as a form of betrayal [8,31,32,75]. Such perceptions of betrayal have been associated with increased psychological distress and mental health problems and have been shown to fuel distrust in government institutions [23,70]. 

Second, *psychological and motivational barriers* formed a substantial cluster of identified barriers. These included a sense of having already done enough (46%, 463/1006), low motivation or interest (3%), feelings of hopelessness (‘it’s too late’, 3%), denial (1%), perceived limited impact of individual actions (4%), and reluctance to give up one’s lifestyle (4%). A smaller number referred to social comparison and perceived inaction by others (2%), or a lack of power and voice (1%). These barriers reflect emotional distancing, defensive coping, and perceived inefficacy, which have been described in the literature as common responses to climate anxiety and perceived threat (e.g., [11,58]. 

A third group of barriers related to *practical and knowledge-based obstacles*, including lack of knowledge (8%), time constraints (7%), and perceptions that pro-environmental behaviour requires too much effort (4%) or that taking the first step is difficult (1%). These barriers are more logistical in nature and may be more amenable to targeted interventions such as clear information, education, and practical support. Finally, *social and personal factors* also played a role. Barriers related to the social environment were mentioned by 3% of respondents, while 2% referred to personal wellbeing factors such as health, stress, or low energy levels. These findings underscore the importance of considering both social norms and individual capacities in climate engagement.

Taken together, the findings show and strengthen the idea that young adults face a complex mix of structural, psychological, practical, social, and personal barriers to pro-environmental behaviour. In the Dutch context, structural barriers, including financial constraints, housing issues, and perceived dependence on governmental or corporate action, were particularly prominent, while psychological and motivational factors, such as feelings of having already done enough, hopelessness, and perceived inefficacy, were also common. These patterns are broadly consistent with findings from pan-European surveys that reported similar structural and motivational obstacles among adolescents and young adults in countries such as Germany, France, and Spain (e.g., [76]) and show strong similarities to those identified in a review of qualitative studies [46]. Notably, the relatively high proportion of Dutch respondents reporting that they have ‘already done enough’ may reflect cultural and policy-specific factors, including social norms around sustainability. Overall, these results underscore the importance of addressing barriers at multiple levels—systemic, emotional, and practical—when designing interventions and policies to support youth engagement in climate action across Europe.

### 4.3. Strengths and Limitations

Although this study was exploratory and the qualitative component was necessarily concise, limited to two open-ended questions, a key strength lies in the high level of engagement: all 1006 participants from a representative sample responded. Most engaged thoughtfully and seriously, with many (33%) providing at least one open-ended response. The willingness of many participants to respond despite survey length suggests that climate change is an important issue to them. A major strength of the present study lies in its integrated approach, examining both climate-related emotional responses and the associated emotional and psychological needs within the same sample and at the same time. In contrast, much of the existing literature addresses emotions or needs separately. The qualitative data played a crucial role in interpreting the quantitative findings, while the quantitative results, in turn, helped to contextualise and deepen the understanding of the qualitative insights. However, several limitations should be acknowledged.

First, the analysis was conducted by a single coder and relies on brief open-text responses (maximum three per participant), limiting the depth and contextual richness of the data. Therefore, the thematic findings should be interpreted cautiously and cannot be considered generalizable or as providing a full qualitative insight into the phenomena under study. 

Second, we used a sliding visual analogue scale (0–100) to assess emotional responses. Prior research indicates that VAS scores tend to produce lower means and greater variability compared to Likert scales, potentially affecting the comparability and interpretation of absolute emotional intensity scores. At the same time, the use of a continuous VAS allowed for more fine-grained measurement of emotional intensity, which may capture subtle variations that ordinal Likert scales could miss. 

A notable limitation is the inclusion of two fixed-answer options alongside the open-ended questions. Positioned at the end of a longer questionnaire, respondents may have chosen these fixed options for convenience rather than investing time in open-ended answers, potentially leading to rushed or less thoughtful responses. Additionally, the second question was negatively framed, assuming respondents experienced environmental distress and negative emotions, which may have biased answers. These factors highlight potential data biases and underscore the need for further research using more balanced question formats and response options.

### 4.4. Practical Implications

This study offers exploratory insights into the emotional landscape surrounding climate change and the perceived barriers to pro-environmental behaviour among young adults. While the sample was representative and response rates were good, the study design does not allow for causal interpretations or broad generalisations. Nevertheless, we can outline some tentative directions for future research and public health policy, considering both short- and long-term strategies. 

Future research could build on these findings by using longitudinal and experimental designs to examine how emotional needs and perceived barriers to pro-environmental behaviour interact over time and influence behavioural intentions and actions (see also [35]). Moreover, more extensive qualitative studies or qualitative components in a mixed-methods study could provide a deeper understanding of the lived emotional experiences and contextual factors that underlie these patterns. More research is also needed to explore more deeply how and in which strength environmental change contributes to the general wellbeing of youth and young adults. Young people are facing a complex mixture of worries and anxieties, also in the Dutch context, on housing problems, social difficulties, and financial constraints (e.g., [28]). Further research is also needed on behaviours that exemplify the ‘value–action gap,’ in which young people recognize the environmental impact of actions such as flying or consuming meat, yet feel unable or unwilling to modify these habits [37,41,57]. Expanding research to larger and more diverse populations—including international samples and participants with varying educational and socioeconomic backgrounds—would strengthen the evidence base. At the same time, locally informed studies, such as this study, remain essential for developing tailored interventions that address specific cultural, social, and structural contexts.

The findings of this study highlight that climate-related emotional responses among young adults are diverse and complex. Public health strategies should therefore move beyond one-size-fits-all approaches and recognize both individual and collective needs. In the short term, interventions should normalize and validate climate-related emotions, provide low-threshold emotional support (such as peer support groups, community-based dialogues), and avoid framing that places sole responsibility on individuals. Tailoring interventions to distinct emotional profiles can enhance effectiveness and engagement (see also [30]). Over the longer term, mental wellbeing in the context of climate change should be more closely embedded in public health systems, linking individual wellbeing with structural and collective pathways for action (see also, [77]). Also, the consideration needs to go beyond addressing the mental health impacts of extreme events only but also cover those related to climate anxiety and indirect impacts. Education programs should integrate emotional competencies alongside climate knowledge, fostering resilience, hope, and agency (e.g., [78,79]). When programs explicitly attend to emotional responses and psychosocial dimensions (e.g., processing distress, building hope and agency), youth show better engagement and adaptive responses to climate concerns [79]. Finally, monitoring inequalities and differential vulnerabilities is essential to ensure that support reaches the most at-risk groups, preventing the amplification of existing health disparities. 

The barriers identified among Dutch young adults in engaging with pro-environmental behaviour highlight both individual and systemic challenges that require multi-level public health responses. In the short term, interventions should focus on reducing cognitive and practical obstacles by providing clear, accessible information, educational resources, and tools that facilitate initial steps toward sustainable living (also see [71,80,81]. Our findings align with previous research indicating that financial costs remain a key barrier to sustainable consumption among environmentally motivated young adults [46,82]. As a policy implication, promoting low-consumption experiences over resource-intensive goods and practices—such as fast fashion and frequent air travel—may both enhance well-being and reduce financial burdens. In addition, research suggests that pro-environmental behaviours, including recycling and sustainable dietary choices, become more attractive and widespread when social norms shift and high-climate-impact lifestyles lose social legitimacy, as normative and moral processes can rapidly shape behaviour [83,84,85,86], especially via social media.

Addressing psychological and motivational barriers, such as feelings of hopelessness, defensiveness, or perceptions of having ‘already done enough,’ requires peer-support mechanisms, positive framing, and role modelling to enhance engagement and self-efficacy. Social norms also play a role, and campaigns can leverage collective challenges or visible examples to reinforce pro-environmental behaviour. 

The observation that half of our participants already feel they are doing what they can raises concerns about potential tokenism—that is, the tendency to engage in a limited number of visible, low-effort actions (e.g., turning off taps or lights) and subsequently perceive one’s responsibility as fulfilled [46]. Consistent with this interpretation, a substantial proportion of Italian youth aged 15–35 (>40%) recently reported feeling no social pressure to reduce their environmental footprint, while only 16–19% prioritised ethical or sustainable consumption [87]. At the same time, the widespread perception that individual behaviour is “too little, too late” underscores the need for structural and systemic change, including supportive policies such as subsidies, infrastructure for sustainable mobility [88], housing adaptations, and accessible sustainable consumption options that reduce environmental constraints and help sustain hope [68]. Public health strategies should also clarify the impact of collective and institutional actions, demonstrating how these reinforce individual efforts, and involve children, adolescents and young adults, who together constitute more than one third of the global population but remain underrepresented in climate decision-making processes [89]. Finally, integrating climate action with broader wellbeing initiatives may help ensure that pro-environmental behaviour supports mental health, resilience, and overall quality of life.

Addressing climate-related emotional needs and barriers to pro-environmental behaviour requires shared responsibility across multiple societal levels. Policymakers play a central role in reducing structural and financial barriers, ensuring access to sustainable options, and integrating climate-related mental health into prevention strategies, in line with calls for system-level responses to climate change and mental wellbeing (e.g., [77,89]. Public health, youth care, and educational institutions are essential in monitoring climate distress, providing emotional support, and fostering coping skills and agency. Individuals and communities contribute through personal and collective engagement within the opportunities shaped by these broader social and policy contexts.

## 5. Conclusions

This exploratory study examined emotional responses to climate change and perceived barriers to pro-environmental behaviour among a representative sample of 1006 Dutch young adults (aged 16–35). Although the study was small and explorative in nature, its strength lies in the size and representativeness of the sample, as well as in the thoughtful and extensive responses provided by participants. This indicates both engagement and, for many, a willingness to contribute. The findings reveal a complex emotional landscape in which feelings of worry, frustration, and powerlessness coexist with hope and a generally constructive outlook. Only a small proportion expressed fatalistic views, while for some the issue appeared emotionally distant or was downplayed. Emotional needs were highly diverse and often practical in nature, relating to financial constraints, lack of time, knowledge gaps, or insufficient alternatives. Many participants also located responsibility outside the individual, placing it with governments, large companies, and other countries. Many young adults appeared to be searching for ways to balance environmental action with everyday life and are looking for feasible pathways to contribute.

The findings underscore the need for a comprehensive, multi-level public health response, and future research, that simultaneously addresses emotional needs, structural barriers, and opportunities for meaningful engagement. In this context, lowering barriers to pro-environmental behaviour and fostering supportive environments that enable sustainable action among young adults may enhance wellbeing and strengthen their sense of agency. Public health policy can play a crucial role in this, by addressing knowledge gaps, motivational factors, social norms, structural barriers, and institutional responsibilities, thereby fostering both sustainable behaviour and mental resilience among young adults.

## Figures and Tables

**Figure 1 ijerph-23-00076-f001:**
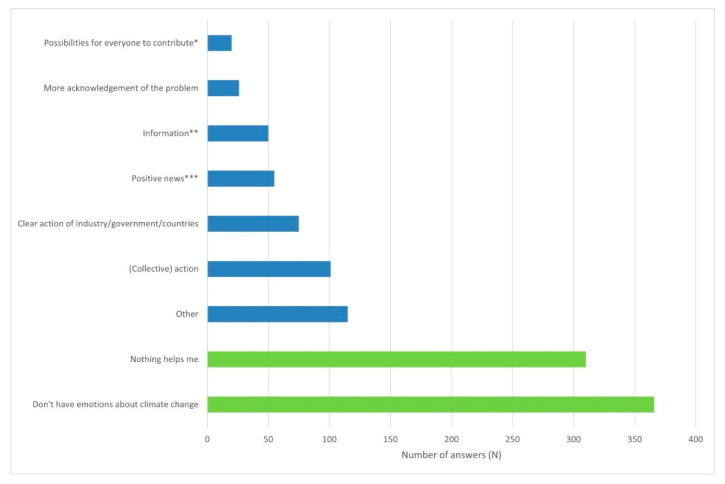
Perceived emotional needs to feel better about climate change, by 1006 Dutch young adults, aged 16–35. Needs are displayed from least mentioned to most frequently mentioned. *Note.* In green: fixed answer option. In blue: open-ended answers. * This included: small clear steps, support in finance and time; and less rules/regulations at the individual level. ** This included: more general attention for the topic, such as in the news; more clear information and examples ‘of what you can do’. *** This included: good news; actual improvements achieved and positive effect of taken actions; ‘see that it works.’.

**Figure 2 ijerph-23-00076-f002:**
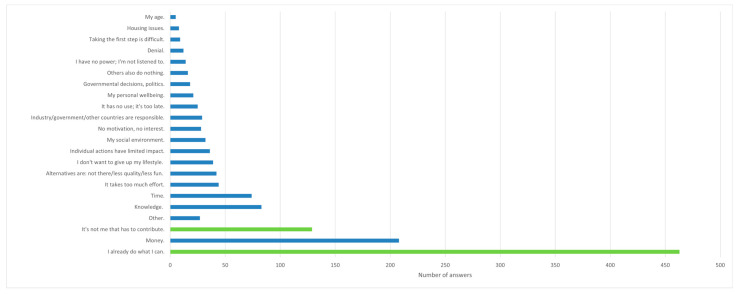
Perceived barriers for pro-environmental behaviour for 1006 surveyed Dutch young adults, listed from less mentioned, to most frequently mentioned answers. Note. In green: fixed-answer option. In blue: open-ended answers.

**Table 1 ijerph-23-00076-t001:** Open-ended questions on emotional and psychological needs in relation to climate change, and perceived barriers to pro-environmental behaviour.

Feeling better. We have asked you several questions about emotions you may experience regarding climate change.			
Question: Can you list up to three things? What could help you feel better about climate change?			
A	1. Write down your answer	2. Write down your answer	3. Write down your answer
B	Not applicable: nothing can make me feel better about climate change.		
C	Not applicable: I experience no emotions related to climate change.		
Sustainable behaviour. We asked you what you are currently doing to live more sustainably and help combat climate change.			
Question: Can you list up to three things? What might prevent you from taking action to live more sustainably yourself?			
A	1. Write down your answer	2. Write down your answer	3. Write down your answer
B	Not applicable: nothing prevents me. (I already try to live sustainably)		
C	Not applicable, I believe it is not my responsibility to contribute.		

Note. Translated to English from the original questionnaire, which was in Dutch.

**Table 2 ijerph-23-00076-t002:** Emotional responses to climate change in Dutch young adults (age 16–35).

	VAS 0–100 [SD]	VAS Median (IQR)	Proportion (Totally) Agree %	Proportion Neutral %	Proportion (Totally) Disagree %
Afraid	22 (24)	11 (3–37)			
Depressed	23 (25)	12 (3–41)			
Stressed	23 (25)	11 (3–40)			
Lonely	20 (24)	8 (2–32)			
1. I worry about the future for next generations, such as (my) children.			57.3	27.7	14.9
2. I feel sadness for the people and animals suffering because of this.			55.0	27.5	17.4
3. I have disagreements and/or arguments with family or friends about how to solve these problems.			15.3	30.4	54.3
4. My health is affected by this.			21.9	33.7	44.2
5. I experience mental health issues (such as anxiety, sadness) because of this			14.6	28.0	57.4
6. I feel I have so little control over it; I feel powerless.			31.2	33.2	35.6
7. I feel a responsibility to point out the consequences of climate change and related problems to others.			28.6	37.2	34.3
8. I often criticize myself for not doing enough to live more sustainably.			22.1	32.7	45.2
9. Large companies should be doing more.			60.7	26.3	13.0
10. I personally do not do enough to live more sustainably.			26.3	37.9	35.9
11. I feel guilty about what my past lifestyle has contributed to this problem.			19.9	32.1	48.0
12. I feel powerless—I cannot change the system that is responsible for this.			35.1	36.4	28.5
13. All the information about what harms the climate and environment overwhelms me. I would rather do nothing			17.5	38.6	43.9
14. I feel guilty about my existence and the effects I have on the planet (e.g., all the waste I produce).			16.0	32.1	51.9
15. I am working on changing my behaviour and lifestyle.			45.6	37.1	17.4
16. I confront others about living more sustainably, even if this leads to conflict.			22.6	28.3	49.1
17. I see it as a challenge and focus on the things I can do.			41.9	34.2	24.0
18. I believe nothing can be done anymore; it is already too late.			18.0	32.8	49.2
19. I avoid the topic (e.g., news about climate change) and focus on things I enjoy.			26.3	36.2	37.5
20. I find support on these topics from peers and/or a (climate) organization.			19.7	38.2	42.1

Note. The table presents anxiety, depression, stress and loneliness (VAS scale 0–100); followed by 5-Likert scale items anxiety (question 1–6), guilt (question 7–14) and coping (question 15–20). 1–12: N = 1003; 13–32: N = 1006.

## Data Availability

The original questionnaire and a translated English version, including the research protocol and supplements from previous related studies, can be accessed online at OSF: https://doi.org/10.17605/OSF.IO/CYGBW (accessed on 15 October 2024). Research data are available from the first author upon reasonable request.

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
