# Peer review of "Emotional Needs in the Face of Climate Change and Barriers for Pro-Environmental Behaviour in Dutch Young Adults: A Qualitative Exploration"

_ijerph, 2026, doi:10.3390/ijerph23010076_

Round 1
Reviewer 1 Report
Comments and Suggestions for Authors
The paper deals with a timely and essential research question. Understanding how young people experience climate change and their relative feelings is essential to developing public policies that help alleviate mental challenges among this population and also help guide collective action. Below I recommend several minor issues:
- The thematic analysis may help create a narrative and add to the quantitative survey. However, the paper underdevelops this section significantly. Although in the introduction and methods the thematic analysis seems to appear as something essential, later the authors argue that it is only complementary (only two-opend ended questions). The paper should incorporate a more thorough discussion of the thematic analysis results. What emergent themes require the creation or validation of new instruments?
- The discussion about the next steps is currently underdeveloped. I understand that the goal of writing an exploratory paper necessitates caution when making claims. However, more discussion on public policies and also how further research could add to the current knowledge on the topic is essential.
- There is little discussion and comparison of the results with other countries in Europe. Improve the discussion on how to place these results in the literature.
Reviewer 2 Report
Comments and Suggestions for Authors
Dear author(s);
First, I would like to congratulate you on this informative and highly descriptive paper. It describes clearly the emotional needs in the face of Climate Change and barriers for pro-environmental behaviour in Dutch young adults. I have four major concerns however:
- The theoretical foundation/basis is underdeveloped. You need to split the Introduction section and provide a clear review of relevant literature and engage in more in-depth analysis of the existing knowledge and literature in this field, in addition to typical patterns of behaviours that are widely observed in other close contexts, perhaps the larger European context.
- You did not discuss clearly what types of environmental challenges are the most influential/distressing ones (e.g. crises, disasters, major incidents, etc...).
- Your method is limited and does not provide rich information. I agree that the major questions of the study were answered, yet they are limited. You can benefit more from your findings and discuss other highly-related issues of equal or even greater influence/importance.
- In the discussion section, you simply discussed the your findings without presenting what short-term and long-term policies should be developed in response to these findings. It is not enough to describe the case without drawing clear policies and reflections on current practices and government directions. These reflections need to be taken into consideration at individual, family, nationwide, and government levels. In other words, you need to specify the roles and responsibilities of each of these groups in response to environmental challenges.
Reviewer 3 Report
Comments and Suggestions for Authors
The data presented, a large and representative sample of responses to two open-ended questions, are valuable and deserve to be reported. However, I think the paper could be rewritten to discuss those data more effectively. My biggest concern is whether this paper is appropriate for this journal. It is only marginally related to the idea of public health. The opening statement about public health implications makes the assumption that helping young people to engage in pro-environmental behavior will be good for the health, without providing clear support for this assumption. Perhaps because of the attempt to make public health claims without fully discussing them, the paper lacks some focus. Is it primarily concerned with mental health, or with encouraging sustainable behavior? In addition, there is some slipperiness about whether they are concerned with emotions and attitudes about climate change in particular or about environmental problems more generally.
The discussion could go more deeply into the meaning of the results. In particular, the results reported in Table 2 are not really discussed, nor related to the open-ended responses. The barriers are better discussed than the emotional needs.
The implications of the results for strategies to encourage behavioral change are useful.
Comments on the Quality of English LanguageThe language was grammatical, but there were occasionally nuances of meaning that appeared to be missing or incorrect. For example:
Line 475 implies that perspective taking is the same as cognitive distancing, whereas this is not how I understand these terms. “Perspective taking” typically refers to trying to take another person’s point of view.
Line 494 implies that despair is a form of coping or emotional distance. Despair , which I understand as a strongly negative emotion, doesn’t sound like either of those things to me.
Line 547 refers to “bottleneck” behaviors, but again this is not the meaning I would associate with bottleneck. A bottleneck is something you have to get through in order to engage in other behaviors.
Round 2
Reviewer 1 Report
Comments and Suggestions for Authors
The revision is satisfactory.
Reviewer 2 Report
Comments and Suggestions for Authors
Dear author(s);
I can see that the document has been carefully revised and all the necessary corrections have been made. Congratulations.